# GAUSSIAN MOTION FIELD FOR HIGH-PERFORMANCE VIDEO COMPRESSION

## ABSTRACT

Neural video representations have advanced video compression technologies, yet many remain decoding-heavy and struggle to model high-frequency motion. To this end, we introduce Gaussian Motion Field (GMF), a 2D Gaussian–Splatting video codec that represents each frame with a compact set of Gaussians updated by a learned motion field. By predicting per-Gaussian deformations for temporal interpolation, GMF reduces temporal redundancy and requires substantially less capacity than traditional methods that rely on keyframe compression and complex motion estimation. In contrast to NeRV-style models with deep convolutional up-sampling, GMF integrates shallow MLPs with lightweight Gaussian representations for efficient decoding. This design yields high storage efficiency and extremely fast decoding: over **1,000 FPS** on a single GPU, amounting to roughly a **50× speedup** over recent methods such as HiNeRV, while maintaining comparable visual quality.

## 1 INTRODUCTION

Video compression encompasses a suite of techniques aimed at reducing the size of video data for more efficient storage and transmission (Ma et al., 2019). Its importance has surged with the proliferation of ultra-high-resolution formats (e.g., 4K, 8K) and the dominance of video-centric platforms like streaming services, which generate massive data volumes. Effective compression is therefore indispensable for efficient video storage and transmission, ensuring high visual fidelity during playback, especially under constrained bandwidth conditions.

Traditional video compression standards, such as MPEG-2 (Aramvith & Sun, 2000), H.264/AVC (Wiegand et al., 2003), HEVC (Sullivan et al., 2012), and the latest VVC (Bross et al., 2021), have established a successful hybrid coding framework. This paradigm tackles spatial redundancy within individual frames using techniques like intra-prediction based on neighboring reconstructed pixels, discrete cosine transform (DCT) or similar transforms to concentrate energy, quantization to reduce precision, and entropy coding (e.g., Huffman or arithmetic coding) to represent symbols efficiently. Temporal redundancy across frames is managed through inter-prediction, which involves estimating motion between blocks in the current frame and reference frames (previously coded frames) and transmitting only the motion information and the prediction residual. Successive

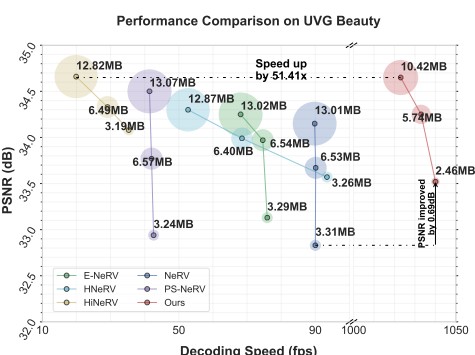

Figure 1: **Quantitative comparison of different video compression methods**.

standards like HEVC Sze et al. (2014) and VVC Bross et al. (2021) have progressively enhanced this framework by incorporating more sophisticated tools, including hierarchical coding structures (e.g., quadtree plus multi-type tree partitioning), a wider array of intra and inter prediction modes, advanced in-loop filtering techniques (like deblocking filters and sample adaptive offset) to reduce coding artifacts, and improved entropy coding schemes. Despite their remarkable efficiency, these hand-engineered pipelines face inherent limitations in modeling complex, non-linear textures and motion patterns, often necessitating a difficult balance between compression rate, perceptual quality, and the computational demands of encoding and decoding. Moreover, the predictive dependency between frames mandates a sequential decoding process.

Figure 2: **Illustration of different video compression methods**.

In contrast, recent progress in implicit neural representations (INRs), which model signals as continuous functions mapping coordinates to target values, have opened new directions for video compression, building on their success in novel view synthesis (Mildenhall et al., 2020). By representing videos as functions that map temporal coordinates $t$ to corresponding pixel intensities $I(t)$, INRs enable continuous and smooth reconstruction of frames while significantly reducing storage overhead. Unlike traditional methods that require sequential decoding, INRs support random-access decoding at arbitrary time indices, facilitating highly parallelized decoding and enhanced efficiency. Typically, these approaches employ convolutional blocks for upsampling latent features and utilize model compression techniques to reduce storage requirements. However, these models often struggle to capture high-frequency content due to limited capacity and the computational overhead of convolutional operations, which can degrade visual quality.

To overcome the challenges posed by both traditional and recent neural methods, we introduce **Gaussian Motion Field (GMF)** – a novel approach that enables compact, expressive, and efficient representation of video content and dynamics. The key insight of GMF is to *represent each video frame using a set of 2D Gaussians, where each Gaussian encapsulates a localized region, capturing spatial information with minimal redundancy*. Unlike methods relying on complex networks for direct frame synthesis or explicit temporal modeling of Gaussians, we introduce the concept of **Motion Field** that captures temporal evolution as a continuous function over both spatial and temporal domains. This motion field predicts the deformation of each Gaussian based on its spatiotemporal coordinates, thereby implicitly capturing motion and enabling smooth interpolation between key frames. Crucially, rather than employing computationally intensive convolutional operations typical of neural video compression frameworks, GMF leverages lightweight multi-layer perceptrons (MLPs) to predict deformation fields efficiently. This design choice significantly reduces computational overhead and leads to faster decoding, as demonstrated in Fig. 2.

While our motion field approach offers an efficient way to model temporal deformation, accurately capturing the dynamics of real-world video scenes using this continuous representation presents significant challenges. First, abrupt scene transitions often induce sharp content discontinuities, which violate the continuity assumption underlying motion fields and hinder accurate deformation prediction. Second, regional variability, where different regions of a scene exhibit distinct motion patterns (e.g., various moving objects), can complicate the learning process for a single continuous motion field and potentially impede training stability.

To this end, our GMF brings two architectural innovations: 1) To cope with large scene changes, we incorporate key frames and propose bidirectional deformation prediction, allowing the motion field to interpolate Gaussians using forward and backward references (i.e., context), thereby simplifying the deformation prediction task. 2) To manage spatially diverse motions, we decompose the complex dynamics captured by the motion field into several learnable motion primitives, each specializing in a distinct motion pattern. The final deformation is then calculated as a weighted sum of these primitives, with weights predicted using interpolated features derived from the motion planes. This modular design not only stabilizes training but also accelerates convergence.

Our design is inspired by the principles of traditional video coding while embracing the flexibility of continuous neural representations. The advantages of GMF are threefold:

- By encoding each frame as a set of 2D Gaussians, GMF captures strong intra-frame pixel correlations and suppresses spatial redundancy more effectively than pixel- or block-based representations.
- Gaussian-level deformation allows GMF to represent complex spatiotemporal dynamics with greater expressiveness and reduced overhead compared with pixel- or block-level motion prediction.
- Based on a lightweight rendering pipeline, GMF achieves over **1,000 FPS** decoding and is more than **50× faster** than recent codecs such as HiNeRV while preserving competitive visual quality.

## 2 RELATED WORKS

**Traditional Video Compression.** Modern video compression rely on hybrid predictive standards such as MPEG-2(Aramvith & Sun, 2000), H.264/AVC(Wiegand et al., 2003), H.265/HEVC(Sullivan et al., 2012), and VVC(Bross et al., 2021), which curb spatial redundancy through intra-frame transforms and temporal redundancy via motion-compensated prediction. Despite decades of refinement, these systems remain collections of hand-crafted modules—prediction, motion search, quantization—that are tuned in isolation, leaving little room for end-to-end optimization. Deep learning has therefore been introduced to upgrade individual components, from intra prediction(Choi & Bajić, 2019) and motion estimation(Ibrahim et al., 2018) to mode decision(Kuanar et al., 2018), entropy modeling(Ding et al., 2021), and post-processing(Dai et al., 2017). Nevertheless, such integrations merely augment the legacy pipeline; block-based learned compression(Chen et al., 2019) still exhibits partition artifacts, and residual-learning methods (Tsai et al., 2018) remain tailored to specific domains.

**Neural Video Compression.** NeRV (Chen et al., 2021) pioneered an image-wise implicit representation for video by mapping time embeddings to latent features with MLPs and decoding frames through convolutional blocks. This implicit neural representation (INR) framework accelerates decoding and matches the rate–distortion performance of conventional codecs, inspiring a wave of successors (Kwan et al., 2024b; Zhu et al., 2025a). ENeRV (Li et al., 2022) further decomposes spatial and temporal contexts to speed up convergence. HNeRV (Chen et al., 2023) adopts content-adaptive embeddings, improving both reconstruction quality and training stability compared with content-agnostic designs. Incorporating flow cues, FFNeRV (Lee et al., 2023) enhances fidelity while suppressing temporal redundancy. More recently, Yan et al. (2024) introduced sparse, learnable static and dynamic codes that obviate explicit optical-flow or residual supervision.

**Dynamic 4D Gaussian Representation.** 3D Gaussian Splatting (3DGS) (Kerbl et al., 2023) has significantly advanced novel-view synthesis and spurred progress on 4D Gaussian Splatting (4DGS) for dynamic scenes (Yang et al., 2024; Wu et al., 2024; Li et al., 2024; Huang et al., 2024; Fridovich-Keil et al., 2023). Contemporary 4DGS systems deform canonical 3D Gaussians over time via dedicated deformation networks, employing architectures such as MLPs, optimizable control points, hexplane decompositions (Cao & Johnson, 2023), and polynomial parameterizations. While our approach shares the idea of dynamic Gaussian modeling (Wu et al., 2024), it targets a broader set of scenarios with larger motion and appearance variation, motivating specialized motion components and richer motion fields than those typically used in 4D Gaussian frameworks.

We summarize the distinctions among representative video compression approaches in Tab. 1. Unlike implicit representation methods, our approach models dynamics through an explicit motion field, delivering intuitive interpretability.

Table 1: **Comparison of video representations.**

| | Explicit Representations (frame-based) | | Implicit Representations (unified) | | Hybrid Representations (frame-based) |
|---|---|---|---|---|---|
| | Hand-crafted | Learning-based | Pixel-wise | Image-wise | Ours |
| Complexity | High | High | Medium | Fair | **Low** |
| Compression Ratio | Medium | **High** | Low | **High** | **High** |
| Explainability | ✔ | ✔ | ✗ | ✗ | ✔ |
| Frame Interpolation | ✗ | ✗ | ✔ | ✔ | ✔ |
| Highlights | H.264, HEVC | DVC | NeRV | NeRF | - |

## 3 METHODOLOGY

### 3.1 OVERVIEW

We observe that redundancy in videos manifests in two forms: *Spatial Redundancy* and *Temporal Redundancy*. On one hand, nearby pixels within a frame are highly correlated, on the other hand, consecutive frames in a video are also highly correlated; these high correlations contribute to video redundancy. Motivated by these observations, we propose the Gaussian Motion Field (GMF) for high-efficiency video representation, illustrated in Fig. 3. GMF aims to reduce both spatial and temporal redundancies by representing frames using a 2D Gaussian model and modeling temporal dynamics through a motion field. After training, the model can be further compressed by pruning unimportant regions of the Gaussians and the motion field.

Figure 3: **Overview of our Gaussian Motion Field (GMF).** Our method is composed of three components. We first use 2D Gaussian representation to model the static content of the frame. Then, we use motion field to predict bidirectional deformation from two key frames and render the target frame with blend rendering. Finally, we use model compression to compress the storage size.

### 3.2 2D GAUSSIAN REPRESENTATION FOR FRAMES

**Preliminary: 3D Gaussian Representation.** 3D Gaussian Splatting enables high-fidelity novel view synthesis by encoding each local region with a Gaussian parameterized by center $\boldsymbol{\mu} \in \mathbb{R}^3$, color $\boldsymbol{c} \in \mathbb{R}^3$, opacity $o \in [0, 1]$, and covariance $\boldsymbol{\Sigma} = \boldsymbol{R}\boldsymbol{S}\boldsymbol{S}^\top\boldsymbol{R}^\top$, where $\boldsymbol{R}, \boldsymbol{S} \in \mathbb{R}^{3\times3}$ ensure positive semi-definiteness. Rendering projects the Gaussians to image space, sorts them by depth, and evaluates each pixel using standard $\alpha$-compositing

$$\alpha_i = o_i \exp\left(-\tfrac{1}{2}(\boldsymbol{p}' - \boldsymbol{x})^\top \boldsymbol{\Sigma}^{-1}(\boldsymbol{p}' - \boldsymbol{x})\right), \quad T_i = \prod_{j<i}(1 - \alpha_j), \quad C(\boldsymbol{x}) = \sum_{i=1}^{N} T_i \alpha_i \boldsymbol{c}_i.$$

**2D Gaussian Representation.** 2D Gaussians (Zhang et al., 2024) mirror the 3D construction but live entirely in image space: each Gaussian has center $\boldsymbol{\mu} \in \mathbb{R}^2$, color $\boldsymbol{c} \in \mathbb{R}^3$, opacity $o \in [0, 1]$, and covariance $\boldsymbol{\Sigma} = \boldsymbol{R}\boldsymbol{S}\boldsymbol{S}^\top\boldsymbol{R}^\top$ with $\boldsymbol{R}, \boldsymbol{S} \in \mathbb{R}^{2\times2}$. This compact representation preserves the blending behavior of the 3D formulation while reducing spatial redundancy in video frames.

The rendering process for image representation differs from that in 3D space. While 3D Gaussian Splatting requires depth sorting for visibility ordering and subsequent alpha blending, 2D Gaussian Splatting eliminates depth ordering and alpha blending, as it operate directly in 2D image space with a single view. Thus, the rendered color $C(\mathbf{x})$ at pixel $\mathbf{x}$ with 2D Gaussian representation is $C(\mathbf{x}) = \sum_{i=1}^{N} \mathbf{c}_i \alpha_i$, where $\alpha_i$ is the alpha value of the $i$-th Gaussian. Representing images with 2D Gaussians have shown potential in several downstream tasks like image representation (Zhu et al., 2025b; Zhang et al., 2025) and video representation (Chen et al., 2025).

### 3.3 MOTION FIELD FOR DYNAMICS MODELING

Recent implicit video codecs often follow the NeRF paradigm (Mildenhall et al., 2020), mapping coordinates through sinusoidal encodings and dense MLPs to synthesize motion. While the positional embeddings enable the model to capture the high-frequency details, the associated MLP parameters grow rapidly with resolution, making the representation ill-suited for lightweight compression.

We therefore introduce a factorized Gaussian motion field that stores motion features on orthogonal feature planes: two time-aware planes $\mathbf{M}_{XT}$ and $\mathbf{M}_{YT}$ and a spatial plane $\mathbf{M}_{XY}$. Factorizing the volume in this way reduces the complexity from $O(N^3)$ voxels to $O(N^2)$ plane elements, yielding a compact structure that still captures directional motion cues amenable to Gaussian rendering.

**Keyframe Selection.** Inspired by conventional video codecs, we monitor motion magnitude and appearance residuals to pick a sparse set of keyframes $\{I_k\}$ that capture abrupt transitions. For every keyframe we learn a Gaussian set $\mathcal{G}_k$ alongside a spatial plane $\mathbf{M}_{XY}^k$.

**Bidirectional Decoding.** Large deformations or newly revealed regions are difficult to extrapolate from a single keyframe, so we decode intermediate frames using bidirectional warping from the adjacent keyframes. For a query time $t$ bracketed by keyframes $I_{k-1}$ and $I_k$, we treat the Gaussian sets $\mathcal{G}_{k-1}$ and $\mathcal{G}_k$ as dual references and sample plane features at their centers $\boldsymbol{\mu}_{k-1} = (x_{k-1}, y_{k-1})$ and $\boldsymbol{\mu}_k = (x_k, y_k)$. Each branch samples motion features from $\mathbf{M}_{XT}$ and $\mathbf{M}_{YT}$, fusing them through an element-wise product to obtain $\mathbf{F}_m$. These motion features are modulated with the keyframe

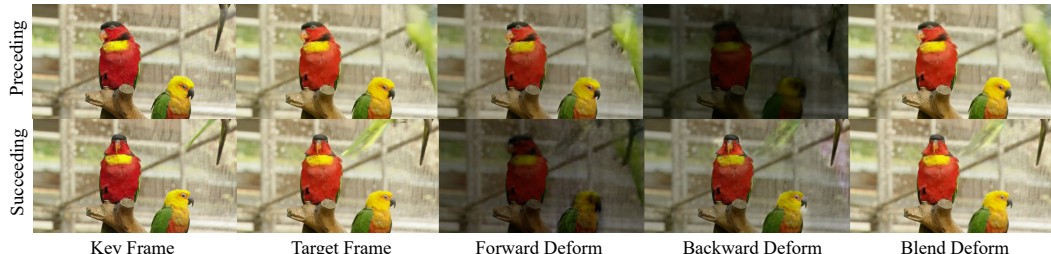

Figure 4: **Illustration of blend rendering**. We show the raw rendering of forward and backward deformation. The brighter part indicates higher influence on the final blend rendering.

appearance codes from $\mathbf{M}_{XY}^{k-1}$ and $\mathbf{M}_{XY}^k$, resulting in composite feature $\mathbf{F}^{\text{fwd}} = \mathbf{F}_m^{k-1} \odot \mathbf{F}_a^{k-1}$ and $\mathbf{F}^{\text{bwd}} = \mathbf{F}_m^k \odot \mathbf{F}_a^k$. The multi-head deformation decoder Wu et al. (2024) reads the composite feature and predicts attribute-specific weights $\mathbf{w}_\pi^{\text{fwd}}, \mathbf{w}_\pi^{\text{bwd}}$ for $\pi \in \{\boldsymbol{\mu}, \mathbf{s}, \boldsymbol{\phi}, \mathbf{c}, o\}$, combining the shared primitive banks $\mathbf{P}_\pi = \{\mathbf{P}_{\pi,1}, \ldots, \mathbf{P}_{\pi,M}\}$. The residual are calculated with $\Delta \boldsymbol{\pi}_t^{\text{fwd}} = \sum_m w_{\pi,m}^{\text{fwd}} \mathbf{P}_{\pi,m}$ and $\Delta \boldsymbol{\pi}_t^{\text{bwd}} = \sum_m w_{\pi,m}^{\text{bwd}} \mathbf{P}_{\pi,m}$, which deform the base Gaussians in $\mathcal{G}_{k-1}$ and $\mathcal{G}_k$.

In practice, we concatenate sinusoidal positional encoding of $(x, y, t)$ to the feature and use separate MLPs for the forward and backward branches so each specializes to its temporal direction.

**Blend Rendering.** Forward and backward deformations can disagree slightly, and without reconciliation their overlap might produce ghosting artifacts. To resolve this, the two branches first produce candidate Gaussians for the same timestep. We spatially associate Gaussians whose centers, scales, and orientations fall within a predefined threshold and interpret their opacities as visibility scores. For each match, we blend attributes with a convex combination weighted by its visibilities (enforcing $\alpha_i^{\text{fwd}} + \alpha_j^{\text{bwd}} = 1$), clamp the results to valid ranges, and discard the duplicate. Unmatched Gaussians are kept with their original attributes. The resulting Gaussian set is used to render the target frame.

In Fig. 4, we present an example of blend rendering, showing the raw rendering results of the Gaussian set after forward and backward deformations. The leftmost column contains two adjacent key frames, and we can see that the corresponding forward and backward deformations remain close to their respective key frames. This indicates that our deformation MLP does not need to learn an overly complex deformation task. Meanwhile, because we use opacity to represent visibility, brighter regions receive higher weight in blend rendering. We observe that the model automatically assigns greater weight to the deformations that align more closely with the target frame.

### 3.4 MODEL COMPRESSION

Capturing complex spatio-temporal dynamics typically demands high-capacity models; to keep the representation compact for storage and transmission we combine progressive training, tile reuse, quantization, and entropy coding.

**Progressive Training.** Jointly optimizing all multi-resolution planes causes redundant features across scales. Following curriculum-style neural rendering (Li et al., 2023; Park et al., 2021), we activate only the coarse levels at the start: with $L_{\text{total}}$ resolution tiers, we train levels up to $L_{\text{start}} = \lfloor L_{\text{total}}/2 \rfloor$ while finer planes remain frozen. Resolutions grow geometrically, $N_l = \omega N_{l-1}$ for $l > 0$ with $\omega = 1.6$, and we unfreeze one higher-frequency level every 500 iterations. During this schedule we disable hierarchical interpolation so each plane learns a distinct frequency band; roughly 60% of the training steps stabilize the coarse tiers before higher-frequency planes join. This staged procedure yields low-entropy structure in early planes and reserves later planes for high-frequency residuals, improving downstream compressibility.

**Tile Reuse Compression.** We organize each apperance plane into fixed-size tiles and learn a reuse gate for every keyframe plane. During training the gate predicts whether a tile can be reused from a previously stored reference frame or must be stored. When exporting, we traverse the planes level by level: tiles whose gate exceeds the reuse threshold are replaced by compact index pointers, while only the different tiles are stored and serialized. We apply the same gate to the planes shared by the forward and backward fields, so any identical content is emitted once and later referenced by both directions. This tile-wise mixture of learned reuse and sparse deltas drastically reduces the payload because spatially coherent regions—already aligned by the motion field—share the same underlying texture, leaving only a small set of residual tiles to account for fine details.

**Quantization and Entropy Coding.** To further reduce the model size, we apply weight quantization to the remaining feature elements. We employ Quant-Noise (Stock et al., 2021) during a brief fine-tuning phase after pruning to mitigate the performance degradation potentially caused by quantization. Finally, the quantized weights of the feature planes are compressed using arithmetic coding (Witten et al., 1987), leveraging statistical redundancies to achieve the final compact representation. Further details of the compression pipeline are provided in the supplementary material.

### 3.5 DISCUSSION ON CONTEMPORARY WORKS

Several contemporary studies employ Gaussians to encode video content. Splatter-a-Video(Sun et al., 2024) models each sequence using 3D Gaussians but relies on auxiliary 2D supervision, such as depth and optical flow. In contrast, our approach is trained solely from raw video frames, eliminating the need for external priors. GaussianVideo (Bond et al., 2025) adapts the 3DGS formulation to model video data, it integrates B-spline-based motion trajectories and Neural ODE-driven camera modeling while discarding auxiliary 2D prior losses. Nevertheless, the inherent redundancy of 3DGS in representing images prevents the approach from achieving optimal storage efficiency. GSVC (Wang et al., 2025) instead records per-frame 2D Gaussian representation, an intuitive alternative that nevertheless remains limited to tasks like frame interpolation and does not generalize further.

## 4 EXPERIMENTS

### 4.1 DATASET AND IMPLEMENTATION DETAILS

To evaluate the effectiveness of the proposed method, we benchmarked GMF against six neural video compression methods: NeRV (Chen et al., 2021), E-NeRV (Li et al., 2022), PS-NeRV (Bai et al., 2023), FFNeRV (Lee et al., 2023), HNeRV (Chen et al., 2023) and HiNeRV (Kwan et al., 2024a) on the UVG dataset (Mercat et al., 2020) (7 videos at $1920 \times 1080$ with a total of 3900 frames). The neural video compression methods were configured at three scales to target the S/M/L scales in NeRV (Chen et al., 2021) for the UVG dataset. This comprehensive scaling approach allows for a fair comparison across different model capacities and video complexities. Following Kwan et al. (2024a), we train each video with a single network separately. We report the encoding and decoding speeds in frames per second, measured using an A100 GPU.

The base plane size is set to $64 \times 64$ for all videos with 32 channels, and we set 3 different levels of planes (up to $256 \times 256$) for each video. We train the model for 5,000 iterations. At the beginning of the training, we only activate the first two levels of planes, and then gradually activate the remaining level at 3,000 iterations. The number of motion primitives is contingent on the video content, ranging

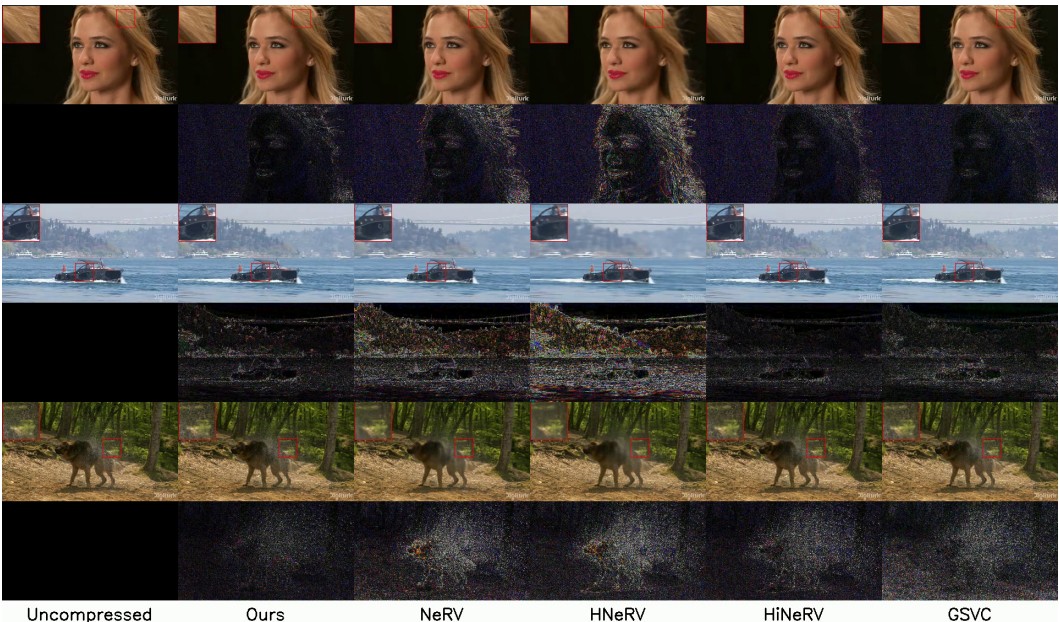

Figure 5: **Qualitative comparison of video compression methods**. The second column of each frame shows the reconstructed error between the original and compressed frames. It can be seen that our method features in capturing high-frequency motion details and fine-grained texture information.

Table 2: **Quantitative comparison of video representation ability on the UVG dataset.** Best results are highlighted in red, second best in orange, and third best in yellow.

| Model | Size | FPS | Beauty | Bosph. | Honey. | Jockey | Ready. | Shake. | Yacht. | Avg. |
|---|---|---|---|---|---|---|---|---|---|---|
| NeRV | 3.31M | 32.4/90.0 | 32.83 | 32.20 | 38.15 | 30.30 | 23.62 | 33.24 | 26.43 | 30.97 |
| E-NeRV | 3.29M | 20.7/75.9 | 33.13 | 33.38 | 38.87 | 30.61 | 24.53 | 34.26 | 26.87 | 31.75 |
| PS-NeRV | 3.24M | 14.7/42.6 | 32.94 | 32.32 | 38.39 | 30.38 | 23.61 | 33.26 | 26.33 | 31.13 |
| HNeRV | 3.26M | 24.6/93.4 | 33.56 | 35.03 | 39.28 | 31.58 | 25.45 | 34.89 | 28.98 | 32.68 |
| FFNeRV | 3.40M | 19.0/49.3 | 33.57 | 35.03 | 38.95 | 31.57 | 25.92 | 34.41 | 28.99 | 32.63 |
| HiNeRV | 3.19M | 10.1/35.5 | 34.08 | 38.68 | 39.71 | 36.10 | 31.53 | 35.85 | 30.95 | 35.27 |
| Ours | 3.86M | 32.0/749 | 35.63 | 38.73 | 39.74 | 37.42 | 31.42 | 37.02 | 33.48 | 36.57 |
| NeRV | 6.53M | 32.0/90.1 | 33.67 | 34.83 | 39.00 | 33.34 | 26.03 | 34.39 | 28.23 | 32.78 |
| E-NeRV | 6.54M | 20.5/74.6 | 33.97 | 35.83 | 39.75 | 33.56 | 26.94 | 35.57 | 28.79 | 33.49 |
| PS-NeRV | 6.57M | 14.6/42.0 | 33.77 | 34.84 | 39.02 | 33.34 | 26.09 | 35.01 | 28.43 | 32.93 |
| HNeRV | 6.40M | 20.1/68.5 | 33.99 | 36.45 | 39.56 | 33.56 | 27.38 | 35.93 | 30.48 | 33.91 |
| FFNeRV | 6.44M | 18.9/49.3 | 33.98 | 36.63 | 39.58 | 33.58 | 27.39 | 35.91 | 30.51 | 33.94 |
| HiNeRV | 6.49M | 8.4/29.1 | 34.33 | 40.37 | 39.81 | 37.93 | 34.54 | 37.04 | 32.94 | 36.71 |
| Ours | 6.65M | 32.5/663 | 35.84 | 38.93 | 39.78 | 37.86 | 33.64 | 39.02 | 32.96 | 36.81 |
| NeRV | 13.01M | 31.7/89.8 | 34.15 | 36.96 | 39.55 | 35.80 | 28.68 | 35.90 | 30.39 | 34.49 |
| E-NeRV | 13.02M | 21.0/68.1 | 34.25 | 37.61 | 39.74 | 35.45 | 29.17 | 36.97 | 30.76 | 34.85 |
| PS-NeRV | 13.07M | 14.1/41.4 | 34.50 | 37.28 | 39.58 | 35.34 | 28.56 | 36.51 | 30.28 | 34.61 |
| HNeRV | 12.87M | 15.6/52.7 | 34.30 | 37.96 | 39.73 | 35.47 | 29.67 | 37.16 | 32.31 | 35.23 |
| FFNeRV | 12.66M | 18.4/49.3 | 34.28 | 38.48 | 39.74 | 36.72 | 30.75 | 37.08 | 32.36 | 35.63 |
| HiNeRV | 12.82M | 5.5/19.9 | 34.66 | 41.83 | 39.95 | 39.01 | 37.32 | 38.19 | 35.20 | 38.02 |
| Ours | 12.57M | 32.1/574 | 35.88 | 40.26 | 39.88 | 38.95 | 35.85 | 40.74 | 33.48 | 37.86 |

from 20 to 50. We also adopt two regularization strategies to stabilize the training process: the total variation loss weighted at 1e-3 and a sparsity loss on primitives with a weight of 1e-4. Additional hyper-parameter details and key frame heuristics are summarized in Appendix A.

## 4.2 VIDEO REPRESENTATION ABILITY

**Frame Reconstruction Results.** Quantitative comparisons presented in Tab. 2 demonstrate the effectiveness of our GMF approach across various metrics. Our method consistently achieves high reconstruction fidelity, comparable to state-of-the-art methods like HiNeRV (Kwan et al., 2024a), while boasting significantly faster decoding speeds. Qualitatively, the benefits of our hybrid representation are evident in Fig. 5. Benefiting from the explicit 2D Gaussian representation for frame content and the implicit motion field for dynamics, our method excels at capturing high-frequency details and fine-grained textures, which can be challenging for purely implicit methods. For instance, in the "Beauty" sequence comparison shown in Fig. 5, our reconstruction preserves sharp facial details accurately, closely matching the ground truth. In contrast, methods like HiNeRV, despite their high capacity often achieved through computationally heavier blocks like depth-wise convolutions, can produce noticeably blurrier results, losing subtle texture information. Similarly, GMF effectively handles challenging motion in sequences like "ShakeIt" (examples shown in Fig. 5), further demonstrating its robustness. This high fidelity is achieved alongside remarkable efficiency. As noted, HiNeRV (Kwan et al., 2024a) increases model capacity at the cost of decoding speed. We also compare the frame reconstruction results with Gaussian-based video compression method, GSVC(Wang et al., 2025), which store frame-wise gaussian representation. GSVC sometimes reconstruct frames with "ellipsoid" artifacts, which is brought by insufficient dynamics modeling. While our method can faithfully reconstruct the dynamics, especially in the "ShakeIt" sequence.

**Frame Interpolation Results.** Our method employs a hybrid video representation, utilizing explicit 2D Gaussians for key frames and an implicit motion field to capture dynamics. This motion field allows for continuous temporal querying, facilitating high-quality frame interpolation between key frames by simply evaluating the model at intermediate time indices $t$.

As demonstrated qualitatively in Fig. 6, our GMF approach generates interpolated frames that exhibit superior visual coherence and smoothness. While Lee et al. (2023) incorporates optical flow guidance, it can struggle with fine details or complex non-linear motion, sometimes leading to artifacts or blurriness in the interpolated frame. Kwan et al. (2024a) may produce reasonable interpolations but can lack the sharpness or detail captured by our method. In contrast, our GMF consistently produces sharp and plausible intermediate frames that closely resemble the ground truth, effectively capturing the temporal evolution between keyframes. This highlights the benefit of our Gaussian deformation field for modeling intricate temporal dynamics.

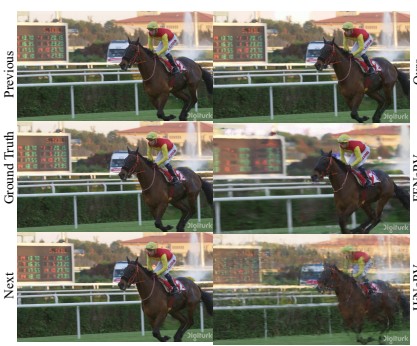

Figure 6: **Frame interpolation results**.

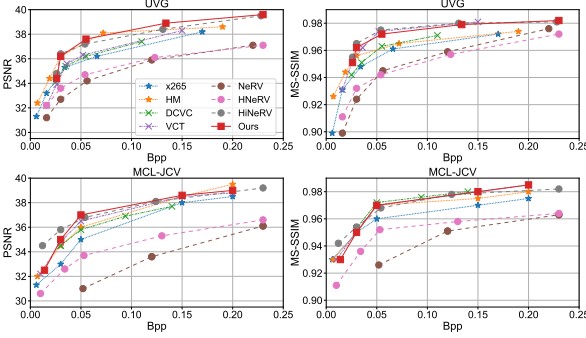

Figure 7: **Results on UVG and MCL datasets**.

## 4.3 VIDEO COMPRESSION RESULTS

We evaluate the rate-distortion (RD) performance of our compressed GMF model against a diverse set of baselines on the UVG (Mercat et al., 2020) and MCL-JCV (Wang et al., 2016) datasets. The baseline methods include: (i) INR-based approaches: NeRV (Chen et al., 2021), HNeRV (Chen et al., 2023), and HiNeRV (Kwan et al., 2024a); (ii) Traditional codecs: HEVC/H.265 (HM reference software) and x265 (medium preset); (iii) State-of-the-art learning-based codecs: DCVC (Li et al., 2021) and VCT (Mentzer et al., 2022). For our method, the actual bitrate (measured in bits per pixel, bpp) is computed by applying arithmetic entropy coding (Witten et al., 1987) to the quantized feature planes after pruning, accounting for the overhead of encoding pruning masks and quantization parameters. Reconstruction quality is measured using PSNR.

Fig. 7 presents the overall RD curves for all methods on both datasets. The results indicate that GMF achieves competitive coding efficiency, particularly when compared to other learning-based and INR-based methods. Notably, GMF significantly outperforms earlier INR methods like NeRV and surpasses established learning-based codecs such as VCT and DCVC across most tested bitrates. While GMF may not reach the peak performance of the highly optimized traditional codecs (HEVC/x265) or the latest top-performing INR models like HiNeRV, especially at higher quality points, its RD performance is strong. Crucially, when considering the substantial decoding speed advantage demonstrated in Table 2, GMF offers a highly compelling trade-off between compression efficiency and practical decoding complexity.

## 4.4 ABLATION STUDY

We conducted ablation studies to investigate the impact of different components of the proposed method. We show the quantitative and qualitative ablation results in Tab. 3 and Fig. 8.

**Effect of Key frames.** We demonstrate the critical role of key frames in the video representation learning. The effectiveness of key frame selection becomes particularly pronounced when video content exhibits significant temporal variations. As shown in Fig. 8, without key frames, the reconstructed frames might contain artifacts, caused by large appearance changes.

Table 3: **Ablation study on MCL dataset**.

| Model | Size | PSNR | MS-SSIM |
|---|---|---|---|
| w/o Multi-level Planes | 12.96M | 33.2 | 0.881 |
| w/o Key Frames | 13.02M | 32.3 | 0.879 |
| w/o Primitives | 13.16M | 31.1 | 0.863 |
| w/o Bidirectional | 13.45M | 35.9 | 0.884 |
| w/o Progressive | 13.56M | 37.0 | 0.896 |
| Full Model | 13.28M | 38.7 | 0.925 |

| Ground Truth | w/o Key Frames | w/o Multi-Level | w/o Bidirectional | Full Model |

Figure 8: **Ablation study**.

**Effect of Multi-level Motion and Appearance Planes.** Our hierarchical decomposition of motion and appearance into multi-level planes reveals substantial performance improvements in handling complex scene dynamics. The coarse-to-fine architecture enables high-fidelity reconstruction of the video content. As shown in Fig. 8, the multi-level motion and appearance planes are able to capture the fine-grained details and the high-frequency information, while without multi-level planes, the reconstructed frames are blurry and fail to capture details.

**Effect of Motion Primitives.** By decomposing complex movements into different motion patterns, these primitives enable compact representation of motions while maintaining temporal coherence. Our ablation shows that primitive-based encoding stabilizes the model training process and improves final performance. Ablating on motion primitives result in failure in fitting specific sequences. Tab. 3 further quantifies this behavior: removing primitives lowers PSNR from 38.7 to 31.1 dB and MS-SSIM from 0.925 to 0.863. The failure manifests early during optimization, where the model overfit static backgrounds yet cannot propagate key frame geometry through time, corroborating the qualitative breakdown shown in Fig. 8.

**Effect of Bidirectional Deformation.** Bidirectional deformation offers two advantages. First, because the target frame blends information from two key frames, the Gaussian count needed for rendering any target frame is at most twice the number used for a single key frame during training, so we avoid storing a large Gaussian set for every key frame. Second, by combining cues from both neighboring key frames it alleviates the difficulty of modeling occlusions and newly appearing objects; while this incurs extra training overhead, it lightens the network workload so we can rely on a more lightweight model. In Tab. 3, dropping the backward branch reduces PSNR by 2.8 dB despite a slightly larger parameter count, highlight-

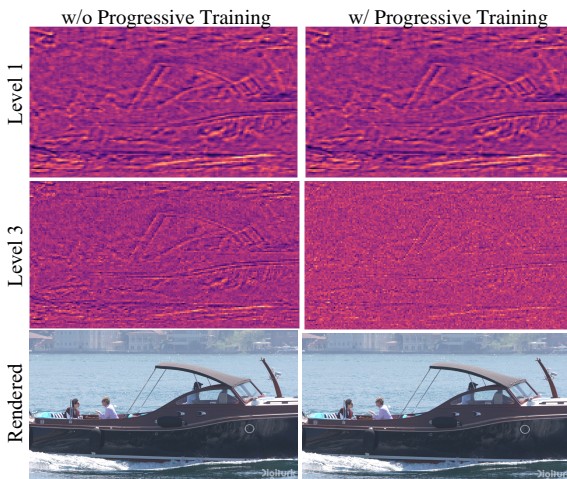

Figure 9: **Ablation study on progressive training**.

ing that contextual fusion matters more than raw capacity. Fig. 8 annotates how the forward-only variant leaves ghost artifacts, and the reconstructed frame fail to capture the background movement.

**Effect of Progressive Training.** Our progressive training strategy, incorporating the late activation of finer feature planes, plays a significant role in structuring the learned representation across multi-resolution motion planes. By introducing higher-resolution planes later during optimization, we encourage them to specialize in capturing higher-frequency motion details and residual information not fully represented by the coarser planes initially activated. This specialization facilitates better information separation across the different plane levels, which is crucial for achieving efficient compression through subsequent pruning and quantization. Quantitatively, disabling late activation drops PSNR by 1.7 dB and degrades MS-SSIM to 0.896 (Tab. 3), indicating that prematurely activating all planes leads to redundant features that are harder to compress. The visualization in Fig. 9 reflects this trend: without progressive scheduling the higher-resolution planes saturate with low-frequency content, while our progressive training keeps coarse planes smooth and reserves finer planes for textured residuals, simplifying subsequent pruning.

## 5 CONCLUSION AND FUTURE WORK

In this work, we presented Gaussian Motion Field (GMF), a video compression method integrating 2D Gaussian representations with an MLP-based motion field. GMF offers a novel approach that addresses limitations found in both traditional video compression methods and recent implicit neural representations. It offers a practical solution for high-quality video compression by reducing spatial and temporal redundancy, providing efficient motion prediction, and achieving significantly faster decoding speed. Although the model already achieves a fairly strong compression ratio, pushing the compression higher leads to significant performance degradation. Improving video reconstruction quality under high compression remains an open research problem.

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

# A ADDITIONAL IMPLEMENTATION DETAILS

This section provides further details on our implementation, complementing Sec. 3 of the main paper. We also provide video results on https://gmfprojectpage.vercel.app/

## A.1 NETWORK ARCHITECTURES

As mentioned in the main paper (Sec 3.3), we employ a multi-head MLP decoder to predict the deformation of Gaussian attributes from the combined feature $\mathbf{F}$. Each head consists of a small MLP with 2 hidden layers, each having 128 hidden dimension, and uses ReLU activation functions. The input to each head is the combined feature $\mathbf{F}$.

## A.2 KEY FRAME SELECTION

Key frames $I_k$ anchor appearance features $\mathbf{M}^k_{XY}$ (main paper, Sec 3.3). We employ a content-adaptive key frame selection strategy, commonly known as adaptive GOP with scene change detection, to enhance compression efficiency. Following the initial key frame (index 0), a subsequent frame $i$ is designated as a key frame if either (1) the minimum GOP interval $G_{\min}$ (e.g., 12) is reached and the SSIM dissimilarity $1 - \text{SSIM}(I_i, I_{i-1})$ exceeds a threshold $\theta_{\text{sc}}$ (e.g., 0.7), indicating a scene change, or (2) the maximum GOP interval $G_{\max}$ (e.g., 120) is reached. Gaussian parameters for new key frames are initialized using the optimized parameters from the preceding frame.

## A.3 MOTION PRIMITIVES

The number of learnable motion primitives $M$ (main paper, Sec 3.3) is chosen based on the complexity of motion in each video sequence. For the datasets used, $M$ typically ranged from 20 to 50, determined empirically during initial experiments for each sequence. The primitives $\mathbf{P}_m \in \mathbb{R}^2$ themselves are initialized randomly with values drawn from a normal distribution $\mathcal{N}(0, 1)$ and are optimized jointly with the rest of the network. No explicit constraints beyond standard weight decay (if used by the optimizer) are applied to the primitives during training.

## A.4 TRAINING DETAILS

We provide additional training hyper parameters complementing those in the main paper (Sec 4.1):

- **Optimizer:** We use the Adam optimizer with $\beta_1 = 0.9$ and $\beta_2 = 0.999$.
- **Learning Rate:** We employ an initial learning rate of 0.001, which decays exponentially over the 5,000 training iterations to a final value of 0.0001.
- **Loss Function:** The primary loss is the L1 reconstruction loss between the rendered frame and the ground truth frame. This is augmented by the regularization losses mentioned in the main paper: Total Variation (TV) loss on the feature planes (weight $10^{-3}$) and a sparsity loss $L_1$ on the predicted primitive weights $w_m$ (weight $10^{-4}$) to encourage sparse activation of primitives. Specifically, the L1 sparsity loss on the primitive weights $w_{i,m}$ for primitive $m$ at Gaussian location $i$ is defined as:

$$L_{\text{sparsity}} = \sum_i \sum_{m=1}^{M} |w_{i,m}| \tag{1}$$

where $M$ is the total number of motion primitives.

The Total Variation (TV) loss is applied to encourage spatial smoothness in the feature planes (e.g., $\mathbf{M}_{XT}$). For a discrete 2D feature plane $\mathbf{F}$, it is typically approximated using finite differences. Specifically, the loss is based on first-order differences:

$$L_{\text{TV}}(\mathbf{F}) = \sum_{x,y} \sqrt{(\mathbf{F}_{x+1,y} - \mathbf{F}_{x,y})^2 + (\mathbf{F}_{x,y+1} - \mathbf{F}_{x,y})^2} \tag{2}$$

where the sum is over all pixel/grid locations $(x, y)$ in the feature plane $\mathbf{F}$. The total regularization loss is a weighted sum of these components.

- **Hardware:** All training and inference speed measurements were performed on NVIDIA an A100 GPU.
- **Training Time:** The averaged training time for each video is about 2 hours, the convergence speed is related to the video content.

## B  MODEL COMPRESSION DETAILS

This section elaborates on the model compression techniques described in Sec. 3.4 of the main paper.

### B.1  QUANTIZATION AND ENTROPY CODING

Following pruning, we apply quantization to the remaining non-zero feature elements in all planes.

- **Quantization:** We use 8-bit quantization for all weights.
- **Quant-Noise Fine-tuning:** To mitigate accuracy loss due to quantization, we employ Quant-Noise during a short fine-tuning phase. This phase lasts for 500 iterations with a reduced learning rate (e.g., $10^{-5}$). Quant-Noise introduces noise during training that simulates quantization effects, making the model more robust to the actual quantization applied post-training.
- **Entropy Coding:** The final quantized weights (and the pruning masks) are compressed using a standard arithmetic coder implementation. We use a simple context model that considers neighboring elements within the feature planes to exploit spatial correlations during coding. The overhead for coding the pruning masks and quantization parameters (min/max ranges for each plane) is included in the final reported bitrate.

## C  THE USE OF LARGE LANGUAGE MODELS

The author acknowledges using a large language model to assist with the writing, especially to correct grammatical errors and polish the article.

