# OpenReview forum: "Gaussian Motion Field for High-Performance Video Compression"
_ICLR.cc/2026/Conference — Submitted to ICLR 2026_

### Official Review · Reviewer_Zgt9 · 2025-10-29

**Soundness:** 2
**Presentation:** 2
**Contribution:** 2
**Rating:** 4
**Confidence:** 4

**Summary:**

This paper proposes Gaussian Motion Field (GMF) for video compression. GMF first learns several key frames using the 2D Gussian Splatting models. Then, it models the motion using a deformation prediction network, which predicts the variation of Gaussian attributes from the bidirectional reference key frames. Finally, 2D Gaussian primitives at arbitrary time index are obtained by applying the predicted deformation. GMF showcases a comparable compression efficiency to HiNeRV and a quite fast decoding speed.

**Strengths:**

1. GMF achieves impressive improvements on the processing speed.
2. The overall idea is simple, and the paper is well-orgnaized.

**Weaknesses:**

1. The rate-distortion performance of GMF has not been compared against state-of-the-art video compression models, e.g., NVRC and DCVC-RT [1]. HiNeRV (published in 2023) is a relatively outdated baseline. It is not convincing enough to achieve a RD performance which is just comparable to HiNeRV.
2. Limited novelty: The proposed GMF seems just a simple extension of 2D Gaussian Splatting representation (GaussianImage) using the 4D reconstruction techniques proposed in 4DGS, lacking inspiring designs for the video compression task.
3. This paper lacks a detailed analysis on the reasons for coding speed improvements. It is important to find out which factor is most important to acceleration. For example, does the acceleration mainly come from the shallow network of GMF, or are the improvements mainly contributed by the fast-rendering property of Gaussian Splatting?
4. Introduction of how to building the Gaussian motion field (Section 3.3) is too brief. Clear definition of variables and technical details are abbreviated, making it difficult to reproduce the method. Please refer to Question 2.

[1] Towards practical real-time neural video compression. CVPR 2025.

**Questions:**

1. Has the time of training the GMF model been counted into the encoding time? Does the FPS in Table 2 correspond to only the frame rendering time? If so, what is the actual encoding/decoding time (i.e., including the time consumption of model compression and entropy coding) of GMF?
2. $F_m$, $F^{\text{fwd}}$, and $F^{\text{bwd}}$ in Section 3.3 (line 215) have not been clearly defined. How to extract these features? How to aggregate them in the network?

---

> ### Author Response · Authors · 2025-12-03
>
> We sincerely thank the reviewer for the thoughtful reading and for recognizing the speed improvements and clear organization.
>
> Q1: Limited novelty.
>
>
>
> A: Thank you for your concern about novelty. Our method differs significantly from 4DGS in two key ways: (1) We target general videos with scene transitions and camera changes, not just well-observed 4D scenes, addressing this through our key frame and blend rendering mechanisms; (2) We focus specifically on video compression with our tile reuse approach that reduces redundancy across key frames. These fundamental differences in problem formulation and technical design make our contribution distinct from simply extending 4DGS to video compression. We hope the reviewer recognizes these meaningful innovations.
>
>
>
> Q2: Analysis on the reasons for coding speed improvements.
>
>
>
> A: The speed gains stem primarily from adopting 2D Gaussian Splatting. When we replace 2DGS with vanilla 3DGS, the decoding speed on UVG drops to roughly 86 FPS, highlighting the efficiency of our 2D formulation.
>
>
>
> Q3: Has the time of training the GMF model been counted into the encoding time? Does the FPS in Table 2 correspond to only the frame rendering time? If so, what is the actual encoding/decoding time (i.e., including the time consumption of model compression and entropy coding) of GMF?
>
>
>
> A: Following the convention in NeRV-style works, the reported encoding time excludes model training. The decoding FPS in Table 2 covers the full decoding process. Because the deformation-prediction network is lightweight, the added cost of model compression and entropy coding is minimal.
>
>
>
> Q4: How to extract motion and appearance features? How to aggregate them in the network?
>
>
>
> A: We use motion planes to encode spatio-temporal cues and feature planes to store spatial information. For a target frame at time t*t*, deformation is predicted from the key frames at $t−1$ and $t+1$, each represented by Gaussians. Motion features $F_m$ are sampled using coordinates $(x,t)$ and $(y,t)$ from the key frames, while spatial features $F^{fwd}$ and $F^{bwd}$ are sampled using $(x,y)$. The features are then fused via Hadamard products.

---

### Official Review · Reviewer_SKpf · 2025-10-29

**Soundness:** 3
**Presentation:** 2
**Contribution:** 3
**Rating:** 6
**Confidence:** 4

**Summary:**

This paper introduces Gaussian Motion Field, a novel neural video compression method that combines 2D Gaussian representations with a lightweight motion field for efficient temporal interpolation. The approach aims to address limitations in both traditional codecs and recent implicit neural video representations by reducing spatial and temporal redundancy through Gaussian-based frame modeling and motion primitives. The authors demonstrate that GMF achieves decoding speeds exceeding 1,000 FPS—roughly 50× faster than recent methods like HiNeRV—while maintaining competitive visual quality and compression performance on standard datasets such as UVG and MCL-JCV.

**Strengths:**

1. Novel Hybrid Representation: The combination of explicit 2D Gaussians for keyframes and an implicit motion field for dynamics is innovative and well-motivated. This hybrid design effectively balances complexity and effectively reduce bitrate
2. Exceptional Decoding Speed: The reported decoding speed of over 1000 FPS is a significant improvement over existing neural video codecs and makes the method highly practical for real-time applications.
3. Efficient Design Choices: The use of factorized motion planes and tile reuse demonstrates a thoughtful approach to model compression and efficiency.

**Weaknesses:**

1. The experimental comparisons could be more comprehensive. While this paper primarily compares traditional codecs and neural representation-based codecs, its comparison with other GS-based video compression methods is not thorough. Particularly, the rate-distortion curves lack experimental results from relevant GS-based approaches (such as [1, 2, 3]). Including these comparisons would more comprehensively demonstrate the performance advantages of the proposed method.
2. The presentation and methodology description in the paper still have room for improvement. For example, how are the 2D Gaussians in keyframes initialized, and how are the relevant initialization parameters selected?


Ref:

[1] GaussianVideo: Efficient Video Representation and Compression by Gaussian Splatting (CVPRW2025)

[2] An Exploration with Entropy Constrained 3D Gaussians for 2D Video Compression (ICLR2025)

[3] GSVC: Efficient Video Representation and Compression Through 2D Gaussian Splatting (NOSSDAV 2025)

**Questions:**

1. Previous works have also utilized mechanisms like MLPs to achieve deformable Gaussians. What are the main differences between this work and those prior approaches?
2. What is the bitrate composition of different components after encoding?
3. In Sec 3.3 Blend Rendering, the authors said "We spatially associate Gaussians whose centers, scales,
 and orientations fall within a predefined threshold and interpret their opacities as visibility scores", could you give a more detailed description of the process?

---

> ### Author Response · Authors · 2025-12-03
>
> Q1: Comparison with other GS-based approaches (such as [1, 2, 3])
>
>
>
> A: We thank the reviewer for pointing out these helpful literatures. GS-based methods typically exhibit on par performance with NeRV. We typically focus on competitive results produced by NeRV-based methods, and we will incorporate these methods in the final version.
>
>
>
> Q2: The initialization for 2D Gaussians.
>
>
>
> A: We explored two initialization strategies: (1) random initialization and (2) seeding with colors sampled from random pixels. Experiments showed that random initialization already performs well and helps avoid getting stuck in local optima.
>
>
>
>
>
> Q3: Previous works have also utilized mechanisms like MLPs to achieve deformable Gaussians. What are the main differences between this work and those prior approaches?
>
>
>
> A: The main differences between our work and previous approaches lie in two key aspects. First, the task settings differ significantly: previous works typically focus on well-observed 4D scenes without scene transitions or substantial camera pose changes, whereas our work targets common video content with diverse dynamics. To address this challenge, we incorporate a key frame mechanism combined with blend rendering. As demonstrated by our experimental results, the proposed blend rendering technique adaptively selects desired areas, effectively mitigating training difficulties.
>
> Second, our work aims to store trained video content with minimal storage requirements. Even though we utilize planes with small dimensions, our tile reuse approach significantly reduces spatial redundancy across planes from different key frames, resulting in more efficient compression.
>
>
>
> Q4: What is the bitrate composition of different components after encoding?
>
> A: The compressed plane, masks and MLPs together with codebook takes about 54%, 25%, 21% storage.
>
>
>
> Q5: In Sec 3.3 Blend Rendering, the authors said "We spatially associate Gaussians whose centers, scales, and orientations fall within a predefined threshold and interpret their opacities as visibility scores", could you give a more detailed description of the process?
>
>
>
> A: The motivation for blend rendering is to mitigate flicker by merging Guassians that represent the same area. In blend rendering proccess, we combine the predicted results from previous and following key frames. For each Gaussian ($G_i$), we define a neighborhood. Inside ($\mathcal{N}i$), the opacities ($\alpha_j$) are normalized into visibility weights
> $
> v_j = \frac{\alpha_j}{\sum{k \in \mathcal{N}_i} \alpha_k}$.These weights blend attributes such as color via $\mathbf{c}i^{\text{blend}} = \sum{j \in \mathcal{N}_i} v_j, \mathbf{c}j$
> The blended tuple replaces the original cluster, preserving occlusion while smoothing out redundant, near-identical Gaussians.

---

### Official Review · Reviewer_KtBi · 2025-11-01

**Soundness:** 3
**Presentation:** 2
**Contribution:** 3
**Rating:** 4
**Confidence:** 5

**Summary:**

In this paper, Gaussian Motion Field (GMF) is proposed for representing video content efficiently. GMF utilizes 2D Gaussians together with a motion field that predicts the deformation of the 2D Gaussians at different time steps. GMF demonstrates fast decoding with high video reconstruction quality and good compression performance.

**Strengths:**

- The method is novel. Applying 2D Gaussians with a lightweight implicit neural representation is an efficient way to represent video but has not yet been studied.
- The results are promising. For example, it achieves performance comparable to SOTA INR methods for video compression, while providing significantly faster decoding.

**Weaknesses:**

- Although GMF shows promising compression performance, the ablation study does not cover the compression techniques such as tile reuse, quantization, and the context model for entropy coding.
- Providing more formal details on the method, for example in the form of equations, would be helpful for understanding.
- The figures for qualitative comparison are low quality. It is hard to compare the proposed method with baselines of similar quality (e.g., HiNeRV).

**Questions:**

- Is the number of training iterations counted in frames or in whole videos?
- Are the results of the baseline methods obtained by your own reproduction, or are they taken from other papers?
- Why do the HiNeRV results in Figure 7 not match those in the original paper? (The lowest-rate point seems to have a bpp much higher than the one reported there.)
- How is the number of parameters controlled for the video representation task?
- How are the Gaussian parameters compressed?
- While the model is optimized only for L1 loss, why is the MS-SSIM performance comparable to, or even better than, INR-based methods that are jointly optimized for L1 and SSIM/MS-SSIM?
- Why is the compression performance of GMF much better than other methods on the Beauty video? A PSNR of 35.63 with only 3.86M parameters is actually much better than many SOTA methods such as VTM. The same observation holds for the Shake video.
- What is the setting of HM?

---

> ### Author Response · Authors · 2025-12-03
>
> We sincerely thank the reviewer for the thoughtful assessment and for highlighting both the novelty of combining 2D Gaussians with a lightweight implicit motion field and the strong reconstruction/compression results.
>
> Q1: Is the number of training iterations counted in frames or in whole videos?
>
> A: Iterations are counted in frames.
>
> Q2: Are the results of the baseline methods obtained by your own reproduction, or taken from other papers?
>
> A: We report the numbers from HiNeRV and follow its configuration for fair comparison. We also reproduced the methods to generate the figures, and the reproduced results match the reported ones closely.
>
> Q3: Why do the HiNeRV results in Figure 7 not match those in the original paper? (The lowest-rate point seems to have a bpp much higher than the one reported there.)
>
> A: Figure 7 uses our reproductions; the bpp shift likely comes from minor configuration differences. On the UVG dataset, our results align closely with the HiNeRV paper.
>
> Q4: How is the number of parameters controlled for the video representation task?
>
> A: We empirically select a configuration that balances quality and efficiency and apply the same setup across videos.
>
> Q5: How are the Gaussian parameters compressed?
>
> A: We compress them using vector quantization.
>
> Q6: While the model is optimized only for L1 loss, why is the MS-SSIM performance comparable to, or better than, INR-based methods that are jointly optimized for L1 and SSIM/MS-SSIM?
>
> A: Thank you for catching this. After rechecking the experiments, we found that we actually use the combined L1 + SSIM loss from Gaussian Splatting. We will correct this description in the final version.
>
> Q7: Why is the compression performance of GMF much better than other methods on the Beauty video? A PSNR of 35.63 with only 3.86M parameters is much better than many SOTA methods such as VTM. The same observation holds for the Shake video.
>
> A: Both “Beauty” and “Shake” contain little camera motion but rich fine-grained dynamics (flowing hair, water droplets). GMF excels at capturing these local motions, so the gains are most pronounced there. For videos with large camera moves or scene transitions, our advantage narrows.
>
> Q8: What is the setting of HM?
>
> A: We use QP values {12, 17, 22, 27, 32}.

---

### Official Review · Reviewer_oJYU · 2025-11-01

**Soundness:** 3
**Presentation:** 3
**Contribution:** 3
**Rating:** 4
**Confidence:** 4

**Summary:**

This paper introduces Gaussian Motion Field (GMF), a novel hybrid approach for efficient video representation and reconstruction. The core idea is to combine the strengths of explicit and implicit representations: using 2D Gaussians as explicit primitives to represent static frame content, and an implicit neural motion field to model the dynamics. The authors evaluate GMF on standard video reconstruction tasks, comparing it against state-of-the-art methods like NeRV, HiNeRV, and others.  Results, both quantitative (e.g., PSNR, bitrate) and qualitative, demonstrate that GMF achieves reconstruction quality on par with or exceeding existing methods.

**Strengths:**

(1) The fusion of explicit 2D Gaussians with an implicit motion field is a creative and well-justified design.  It effectively leverages the compactness and rendering speed of explicit graphics primitives with the flexibility and smoothness of neural implicit functions for motion modeling.

(2) By emphasizing faster decoding speeds, the work addresses a critical practical bottleneck in learned video compression and representation.  This makes the proposed method highly relevant for real-time applications, which is a major contribution beyond just improving reconstruction metrics.

**Weaknesses:**

(1) I think in the experimental section, the author should incorporate a comparison with the latest 3DGS-based method, which is also an implicit representation, to prove the effectiveness of this method.

(2) For Figure 7, I hope the author can provide the BD-Rate and BD-PSNR metrics for quantitative comparison to precisely display the performance improvement values of this method compared to other methods. Furthermore, the author did not make a comparison with the latest video compression SOTA methods (DCVC-FM [1], DCVC-RT [2]).

[1] Li, Jiahao, Bin Li, and Yan Lu. "Neural video compression with feature modulation." Proceedings of the IEEE/CVF Conference on Computer Vision and Pattern Recognition. 2024.

[2] Jia, Zhaoyang, et al. "Towards practical real-time neural video compression." Proceedings of the Computer Vision and Pattern Recognition Conference. 2025.

**Questions:**

Please see Weaknesses.

---

> ### Author Response · Authors · 2025-12-03
>
> We sincerely thank the reviewer for the constructive feedback and for highlighting the strengths of GMF.
>
> Q1: Incorporate comparision with 3DGS-based method.
>
> A: We appreciate the suggestion. We have included the comparsion with GSVC[1] in the qualitative results. Reviewer SKpf also raised several literature about compress video with gaussian splatting, and we will incorporate these results in the final version.
>
>
>
> Q2: BD-Rate and BD-PSNR comparison
>
> A: When taking NeRV as the anchor model:
>
> | Method | BD-Rate(%) | BD-PSNR(dB) |
> | :----- | :--------- | :---------- |
> | NeRV   | 0          | 0           |
> | HiNeRV | -74.098    | 3.082       |
> | Ours   | -74.133    | 3.222       |
>
> Thank you as well for the relevant references; we will incorporate comparisons with those methods in Figure 7.
>
> [1] Longan Wang, Yuang Shi, Wei Tsang Ooi. "GSVC: Efficient Video Representation and Compression Through 2D Gaussian Splatting"

---

### Official Review · Reviewer_W7e7 · 2025-11-01

**Soundness:** 3
**Presentation:** 3
**Contribution:** 2
**Rating:** 6
**Confidence:** 4

**Summary:**

This paper proposes Gaussian Motion Field (GMF), a neural video codec that represents frames using 2D Gaussians and models temporal dynamics through a learned motion field. The method achieves over 1,000 FPS decoding speed (50× faster than HiNeRV) while maintaining comparable visual quality. Key innovations include bidirectional deformation prediction, motion primitive decomposition, and progressive training with tile reuse compression.

**Strengths:**

1. The speedup over HiNeRV is significant and addresses a real bottleneck in neural video compression, making the method more viable for practical deployment.
2. The combination of explicit 2D Gaussians for spatial structure with implicit motion fields for temporal dynamics is intuitive and interesting
3. The paper is well-written with effective visualizations that clearly communicate the technical approach.

**Weaknesses:**

1. On MCL-JCV dataset, GMF significantly underperforms HiNeRV across most bitrates and the PSNR gap widens dramatically as bitrate decreases.
2. The evaluation is insufficient for a video compression paper: Currently evaluated only on: UVG: 7 videos, 1920×1080, primarily static camera scenes and MCL-JCV. Standard practice in video compression research requires evaluation on: HEVC Common Test Conditions (CTC), especially class B~E,  which covers diverse scenarios.
3. The paper claims in Table 2 to report "encoding and decoding speeds" with format "X/Y FPS". However, there are severe contradictions: Paper states: "The averaged training time for each video is about 2 hours" (Appendix A.4), if 32.0 FPS is encoding speed, then for a 600-frame video: 600/32.0 ≈ 18.5 seconds.

**Questions:**

Please refer to weaknesses.

---

> ### Author Response · Authors · 2025-12-03
>
> We sincerely appreciate the reviewers’ thoughtful suggestions and their recognition of our fast rendering speed and methodological innovations.
>
> Q1: On the MCL-JCV dataset, GMF significantly underperforms HiNeRV across most bitrates, and the PSNR gap widens dramatically as bitrate decreases.
>
> A: Our method stores visual information in feature planes and relies on implicit neural networks to decode the resulting deformation patterns. When the bitrate drops below a certain threshold, however, the planes can no longer remain sufficiently compact, which ultimately degrades reconstruction quality.
>
> Q2: HEVC Common Test Conditions (CTC), especially the Class B–E datasets.
>
> A: Because our work primarily benchmarks against NeRV-based methods, we used the same datasets they adopted for fair comparison. That said, our approach is highly scalable and can be applied to other datasets as well. Due to time constraints, we are currently unable to provide additional experimental results for the HEVC CTC datasets, and we hope for your understanding.
>
> Q3: Decoding speed.
>
> A: We follow the encoding-speed convention used by NeRV-based methods, where “encoding time” refers specifically to the time required to encode the model parameters, excluding the training duration. In practice, NeRV-based methods typically need about 4–6 hours to complete training.

---

### Meta-Review · Area_Chair_iqkf · 2026-01-04

**Summary:**

This paper introduces Gaussian Motion Field, a neural video compression framework that integrates 2D Gaussian splatting with a learned motion field to achieve very fast decoding. Reviewers acknowledge the impressive decoding speed and find the overall idea intuitive. However, the contribution is weakened by limited empirical validation and insufficient comparison with prior work. In particular, the experimental results do not convincingly demonstrate competitive compression performance compared to recent methods, which substantially limits the paper’s impact. Overall, the paper should be considered borderline. While the idea is interesting, the current evidence is not sufficient to clearly support acceptance at ICLR, and I therefore recommend reject.

**Reviewer Concerns:**

he main outstanding concerns relate to the experimental evaluation and benchmarking. As highlighted by Reviewer oJYU, the paper lacks comparisons with recent neural video compression methods such as DCVC-FM and DCVC-RT, making it difficult to assess the competitiveness of the proposed approach. Moreover, as noted by Reviewer W7e7, the compression performance remains moderate and degrades noticeably at low bitrates. In addition, Reviewer KtBi points out that the experimental settings are not sufficiently clearly specified, and that the configurations used for the compared methods are not fully aligned with those adopted in this work, raising concerns about the fairness of the comparisons. While the rebuttal clarifies some implementation details and design choices, these clarifications may not be sufficient to address the core concerns.

**Reviewer Scores:**

Based on the rebuttal, the reviewers’ original concerns are unlikely to be sufficiently alleviated to warrant a revision of their scores, and the scores are therefore expected to remain largely unchanged.

---

### Decision · Program_Chairs · 2026-01-26

Reject